# A Bandpass Filter Using Half Mode SIW Structure with Step Impedance Resonator



Min-Hang Weng [1], Chin-Yi Tsai [2], De-Li Chen [1], Yi-Chun Chung [3] and Ru-Yuan Yang [2,*]

1   School of Information Engineering, Putian University, Putian 351100, China; hcwweng@gmail.com (M.-H.W.);
    zhangli_pt@163.com (D.-L.C.)
2   Graduate Institute of Materials Engineering, National Pingtung University of Science and Technology,
    Pingtung County 912, Taiwan; tylertsaiji@gmail.com
3   Bachelor Program in Advanced Materials, National Pingtung University of Science and Technology,
    Pingtung County 912, Taiwan; Taiwanchun871129@gmail.com
*   Correspondence: ryyang@mail.npust.edu.tw; Tel.: +886-8-7703202 (ext. 7555); Fax: +886-8-7740552

**Abstract:** This paper presents a miniaturized bandpass filter, which uses half mode substrate integrated waveguide (HMSIW) structure with embedded step impedance structure (SIS). By embedding the stepped impedance structure into the top metal of the waveguide cavity, the center frequency can be quickly shifted to a lower frequency. The operating center frequency of the proposed bandpass filter (BPF) using HMSIW resonators with embedded SIS is tunable as functions of the parameters of the SIS. The design curve is provided. A filter example of the center frequency of the filter at 3.5 GHz is fabricated and measured, having the insertion loss $|S_{21}|$ less than 3 dB, and the return loss $|S_{11}|$ greater than 10 dB. The transmission zeros are located at 2.95 GHz and 3.95 GHz on both sides of the passband, both of which are lower than 30 dB. The simulation result and the measured response conform to the proposed design concept. The proposed HMSIW filter design is in line with the current 5G communication trend.

**Keywords:** bandpass filter; half mode; substrate integrated waveguide (SIW); step impedance structure

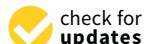



## 1. Introduction

With the advent of the 5G communication era, new communication frequency bands and new communication standards are continuously being announced. In the RF front-end system, a small-sized BPF is responsible for separating the radio signals transmitted and received by mobile phones from different frequency bands [1]. Component miniaturization and module integration are also the current trends in BPF design. Among the filters in the Sub-6 GHz frequency band, low temperature co-fired ceramic (LTCC) filters and surface acoustic wave (SAW) filters are most commonly used in mobile communication devices [2,3]. However, the manufacturing costs of the above filters are still high. In the development of microwave and millimeter-wave filters, the choice of the substrate material and filter structure will affect the transmission loss, which is a very critical design consideration.

Substrate integrated waveguide (SIW) developed in recent years combines the advantages of traditional cavity waveguide and microstrip line [4]. SIW is a synthetic non-planar waveguide that is converted into a planar form by metal perforation, which is made of a pair of periodic metalized through hole arrays or slot trenches, like two parallel fences, so the EM wave has a specific spacing [5]. SIW technology has been applied to the various design and production of innovative filters [6–12]. The SIW filter typically uses more than two SIW resonant modes to form a filter, producing different response characteristics according to the number of the designed SIW resonant cavities [9,10].

Recently, a HMSIW has been proposed with further study in the design of SIW as a new improved technology [13]. When compared to conventional SIW, HMSIW can

reduce the width of the waveguide and the surface area of the metal sheet by nearly half; therefore, has a much simpler component manufacturing process than that of SIW. Several filter components have been reported by using the HMSIW [13–18]. In Zhou's work [15], a compact HMSIW filter was implemented with a dual-mode microstrip resonator. In Huang's work [16], a compact wideband ridge HMSIW filter was reported. A mixed-coupled HMSIW topology was used to provide transmission zeros in filters by mixing the evanescent-mode coupling with the slot coupling. In Chen's work [17], a compact dual-band BPF based on HMSIW resonator with slot perturbation was proposed. The quasi-TEM mode and the $TE_{102}$ mode of the HMSIW resonator were used to provide the dual-band response by adjusting the resonant frequency of the $TE_{102}$ mode with slot line perturbation. Moreover, source-load coupling was also used to achieve high band selectivity. In He's work [18] a compact BPF with the center frequency of 3.5 GHz and the bandwidth of 4.9% was proposed by simply consisting of two HMSIW resonators and the microstrip-line source-load cross-coupling structure. The above BPF design concepts based on the HMSIW all show the advantages of reducing the size of the original symmetrical SIW by half, thereby obtaining a smaller component space. On the other hand, a SIW research trend is gradually forming, that is, a traditional microstrip line structure is formed on the metal surface above the SIW [9,15,17–19]. Thus the entire resonant properties can be combined with the characteristics of the SIW cavity and the microstrip resonators, which can further be used to adjust the frequency and the bandwidth. For example, E-shaped slot lines have been etched on the edge sides of the SIW cavities to couple with magnetic fields of the resonant modes, and thus the center frequencies of the passbands would be flexibly controlled [19]. Microstrip structure is easier to implement on the top metal surface of the HMSIW cavity to tune the filter performance.

In this paper, a BPF, which uses HMSIW structure with the combing of SIS is reported. The purpose of this article is to prove that introducing a microstrip type of SIS on the surface metal of the waveguide can effectively reduce the frequency of operation, thus tuning the resonant frequency to the desired passband without degrading the filter performance. The proposed filter is smaller than its conventional SIW filter due to the use of half mode as well as the SIS and has the advantages of high-frequency selectivity and tuning ability. Moreover, through the proposed design curve, filter design in a certain frequency range becomes simple. As long as the BPF using HMSIW resonators without embedded SIS is designed at the beginning, there will be many options for center frequency adjustment after embedding the SIS. The design concept of the proposed miniaturized BPF will be described.

## 2. Design Procedure

Figure 1 shows the structure of a BPF combined by two HMSIW resonators with the SIS introducing on the surface metal of the HMSIW cavity and a pair of microstrip coupling feed line. Various SIS parameters on the surface metal of the HMSIW cavity were simulated to discuss the design concept of easily tuning the band frequency of the BPF. Duroid 5880 substrates with a dielectric constant of 2.2, a thickness of 0.787 mm, and a loss tangent of 0.0009 were used in this design.

### 2.1. A BPF Using HMSIW Resonators without SIS

Initially, the 0.6 mm diameter of the vias and the vias distance of 0.8 mm were used to satisfy the requirements of the SIW structure [5]. When using a half-width of 11.55 mm, the HWSIW resonator was formed. For a BPF, two HWSIW resonators were coupled and speared with a space (s) [18]. A pair of microstrip coupling feed line with a 78 Ω and 1.11 mm width was used for signal input and output at both ends of the HMSIW resonator. Figure 2 shows (a) structure and (b) simulated filter performance of the proposed BPF using HMSIW resonators without embedded SIS. The equivalent circuit model of the HMSIW filter without embedded SISR was discussed [18], and thus would not be described again here. It was known the length, $L_3$, mainly dominated the center frequency of the BPF response and the metallic vias coupling structure (s) causing the magnetic

coupling provided the suitable coupling coefficient for the two HMSIW resonators without embedded SIS. Based on the above discussion, the center frequency of the proposed BPF was determined by the area ($L_3$) of the original HMSIW resonator first, and then the bandwidth of the proposed BPF was fine-tuned by the space (s) between the two HMSIW resonators. When $L_3$ = 11 mm and s = 1.4 mm, the filter had the center frequency of 4.35 GHz. It was found that two transmission zeros are provided near the passband which is simply resulted from the cross-coupling due to source-load coupling of the feed line [18,20].

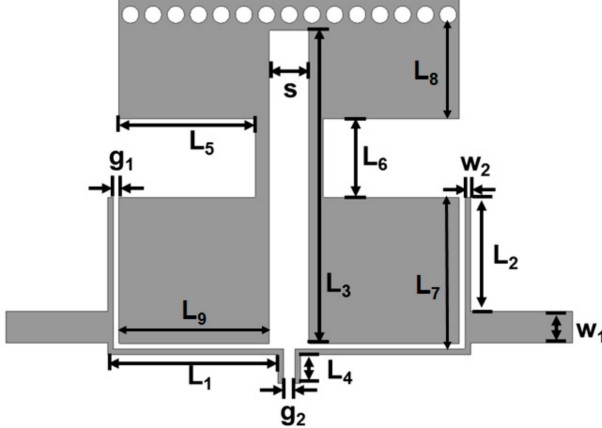

**Figure 1.** Structure of the proposed bandpass filter using half-mode substrate integrated waveguide (HMSIW) resonators with embedded step impedance structure (SIS).

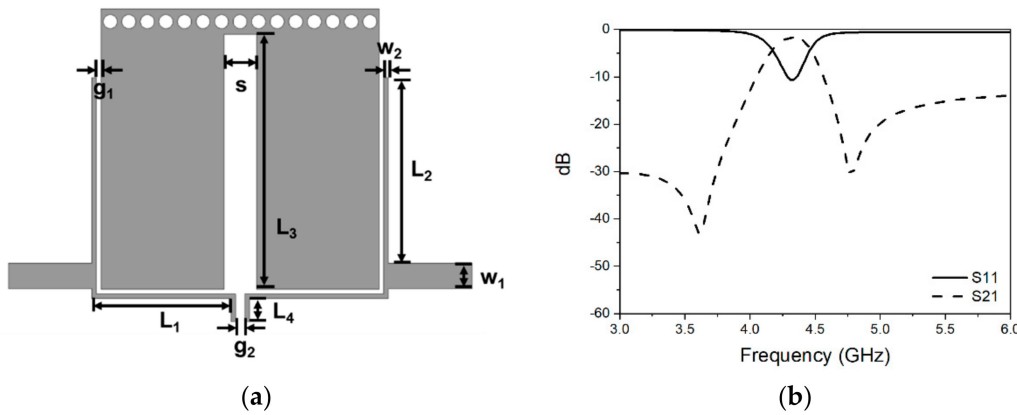

(**a**)                                                                                                 (**b**)

**Figure 2.** (**a**) the prototype structure and (**b**) simulated filter performance of the proposed bandpass filter (BPF) using HMSIW resonators without step impedance resonator (SIR). ($L_1$ = 6, $L_2$ = 4, $L_3$ = 11, $L_4$ = 1, $L_5$ = 0, $L_6$ = 0, $g_1$ = 0.2, $g_2$ = 0.4, s = 1.4, $w_1$ = 1.11, $w_2$ = 0.2, mm in all).

### 2.2. BPF Using HMSIW Resonators with Embedded SIS.

As discussed in [19], E-shaped slot lines were embedded on the top metal surface to adjust the resonant modes of the SIW cavity. The equivalent circuit of the E-shaped slot lines was discussed to expect the effect of resonant modes of the slot lines on the SIW cavity. It is well known that a step impedance resonator (SIR) can be used to shift the resonant modes to a higher or lower frequency. Due to the half mode, the microstrip structure was easy to combine on the top metal surface of the SIW cavity to tune the filter performance. In this study, the microstrip SIS was introduced and embedded on the top metal surface of the HMSIW resonators, as shown in Figure 1. The equivalent circuit of the microstrip step impedance structure could be derived as shown in Figure 3. The physical lengths ($L_7$, $L_6$, and $L_8$), also expressed as the electrical lengths ($\theta_1$, $\theta_2$, $\theta_3$), and the impedances of two different sections related to the widths of $L_9$ and ($L_9$-$L_5$) could be set as $Z_1$ and $Z_s$, respectively. Indeed, $L_7$ was equal to $L_2$ + $W_1$, and $L_8$ was equal to $L_3$-$L_6$-$L_7$. Thus, by

using the transmission line theory [21], the input admittance of the $Z_{in1}$, $Z_{in2}$, and $Z_{in3}$ could be driven, as shown in Equations (1)–(3), respectively. The resonant conditions of the microstrip step impedance structure were set as $Y_{in3} = 1/Z_{in3} = 0$. The parameters of the $L_5$ and $L_6$ related to Zs and $\theta_2$ varied to tune the resonant conditions of the microstrip step impedance structure. Thus, the entire resonant properties could be determined by the characteristics of the HMSIW cavity and the embedded SIS.

$$Z_{in1} = jZ_1 \tan \theta_1 \tag{1}$$

$$Z_{in2} = Z_s \frac{Z_{in1} + jZ_s \tan \theta_s}{Z_s + jZ_{in1} \tan \theta_s} \tag{2}$$

$$Z_{in3} = Z_1 \frac{Z_{in2} + jZ_1 \tan \theta_2}{Z_1 + jZ_{in} \tan \theta_2} \tag{3}$$

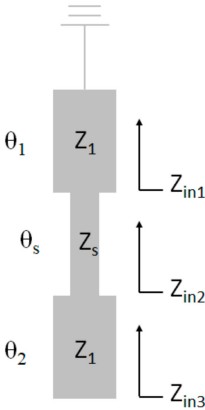

**Figure 3.** Equivalent circuit of the microstrip step impedance structure

Figure 4 shows the effect of $L_5$ on the simulated filter performance of the proposed BPF using two HMSIW resonators with embedded SIS. By introducing the SIS keeping $L_6$ = 2.77 mm and having $L_5$ from 4.4 mm to 5.0 mm, namely, increasing the Zs, the center frequency could be shifted from 4.35 GHz to a lower frequency around 3.0 GHz, indicating that it has the function of frequency tuning.

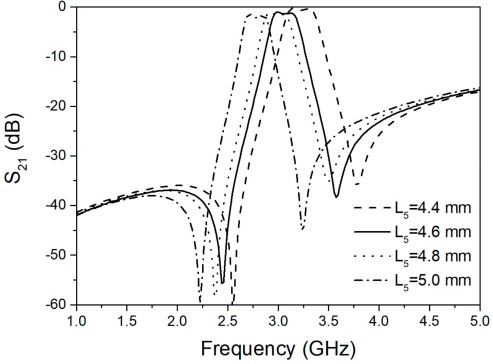

**Figure 4.** Simulated filter performance of the proposed BPF using HMSIW resonators with embedded SIS. ($L_1$ = 6, $L_2$ = 4, $L_3$ = 11, $L_4$ = 1, $L_6$ = 2.77, $L_7$ = 5.11, $L_9$ = 5.3, $g_1$ = 0.2, $g_2$ = 0.4, s = 1.4, $w_1$ = 1.11, $w_2$ = 0.2, mm in all).

Similarly, Figure 5 shows the effect of $L_6$ on the simulated filter performance of the proposed BPF using two HMSIW resonators with embedded SIS. By introducing the SIS keeping $L_5$ = 4.6 mm and having $L_6$ from 1.77 mm to 4.77 mm, namely, increasing the $\theta_2$, the center frequency could be shifted from 4.35 GHz to around 2.8 GHz, also verifying the function of frequency tuning.

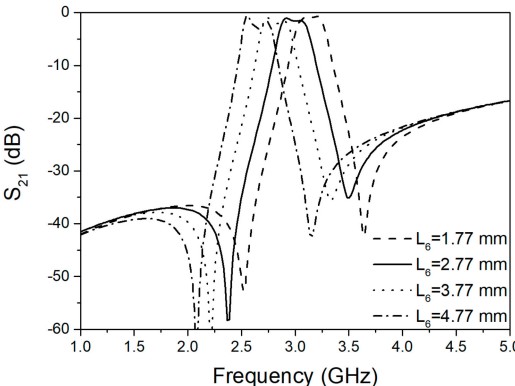

**Figure 5.** Effect of $L_6$ on simulated filter performance of the proposed BPF using HMSIW resonators with embedded SIS. ($L_1 = 6$, $L_2 = 4$, $L_3 = 11$, $L_4 = 1$, $L_5 = 4.6$, $L_7 = 5.11$, $L_9 = 5.3$, $g_1 = 0.2$, $g_2 = 0.4$, s = 1.4, $w_1 = 1.11$, $w_2 = 0.2$, mm in all).

Figure 6 shows the design curve of the center frequency of the proposed BPF using HMSIW resonators with embedded SIS as functions of the $L_5$ and $L_6$. The result indicated the SIS introduced on the surface metal of the proposed HMSIW resonator could quickly reduce the operating center frequency of the designed BPF, when compared to the designed BPF using HMSIW resonators without embedded SIS. It also verified that removal of some part of the top surface metal seems to be equal to the reduction of the cavity size, thus increasing the operating frequency of the SIW cavity. The reduction ratio of the circuit size depended on the parameters of the embedded SIS. For example, in the same circuit size of 12 mm × 21 mm the BPF using HMSIW resonators designed at the same RO 5880 substrate, the BPFs without embedded SIS ($L_5 = 0$ mm and $L_6 = 0$ mm in Figure 1) and with embedded SIS ($L_5 = 4.6$ mm and $L_6 = 2.77$ mm in Figure 1) were operated at the center frequency of 4.25 GHz and 3.25 GHz, indicating a guided wavelength ($\lambda g$) of 54 mm and 71 mm, respectively. Namely, the circuit size of the BPFs without and with embedded SIS were about 0.22 $\lambda g$ × 0.39 $\lambda g$ and 0.17 $\lambda g$ × 0.30 $\lambda g$, respectively. Thus, the BPF using HMSIW resonators with embedded SIS was slightly reduced, when compared to the BPF using HMSIW resonators without embedded SIS. Therefore, when the BPF using HMSIW resonators without embedded SIS was designed at 3.25 GHz, it was expected that the larger circuit size should be used. It was also found that through the proposed design curve, filter design in a certain frequency range became simple. If the BPF using HMSIW resonators without embedded SIS was selected at the beginning, many options for center frequency adjustment could be obtained after embedding the SIS. For example, to meet the 3.8 GHz frequency requirement of Sub-6 GHz for the 5G communication, there would be many design parameter combinations of ($L_5$ and $L_6$) shown in Figure 6 to be selected.

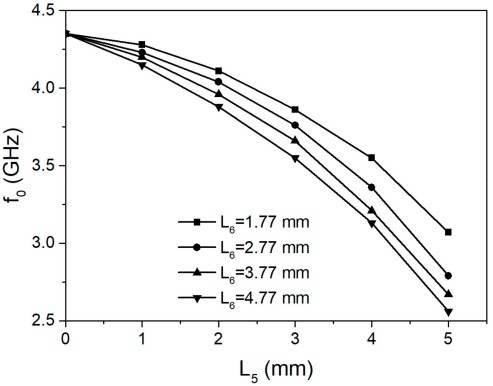

**Figure 6.** Design curve of center frequency of the proposed BPF using HMSIW resonators with embedded SIS as functions of the $L_5$ and $L_6$. ($L_1 = 6$, $L_2 = 4$, $L_3 = 11$, $L_4 = 1$, $L_7 = 5.11$, $L_9 = 5.3$, $g_1 = 0.2$, $g_2 = 0.4$, s = 1.4, $w_1 = 1.11$, $w_2 = 0.2$, mm in all).

Figure 7 shows the simulated filter performance of the proposed BPF using HMSIW resonators with embedded SIS with different coupling space (s). The coupling space (s) was selected as 1.4 mm to have the desired coupling energy without resonant mode splitting.

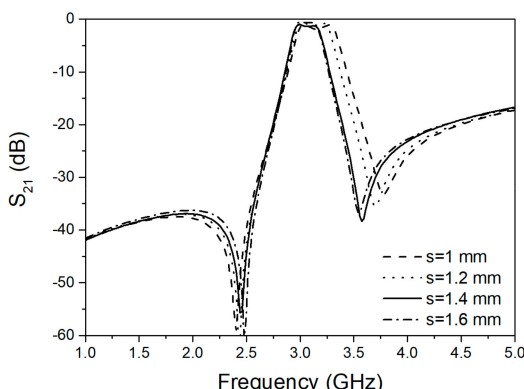

**Figure 7.** Simulated filter performance of the proposed BPF using HMSIW resonators with embedded SIS with different coupling space (s). ($L_1$ = 6, $L_2$ = 4, $L_3$ = 11, $L_4$ = 1, $L_5$ = 4.6, $L_6$ = 2.77, $L_7$ = 5.11, $L_9$ = 5.3, $g_1$ = 0.2, $g_2$ = 0.4, $w_1$ = 1.11, $w_2$ = 0.2, mm in all).

### 3. Experimental Results

Based on the above discussion and the design curve of Figure 6, a filter example was designed at 3.5 GHz to verify the design concept. Moreover, since the line impedance of the pair of microstrip coupling feed lines was very high impedance, the input and output ports were designed as the tapered structure with 78 Ω connected to the coupling feed line and with 50 Ω connected to the network analyzer. Figure 8 shows (a) structure, (b) picture of the fabricated sample, and (c) simulated and measured results of the designed BPF example using HMSIW resonators with embedded SIS. The fabricated BPF was then measured using the network analyzer E5071C. Figure 8a shows the picture of the sample, and its overall size was 12 mm × 21 mm (about 0.17 λg × 0.30 λg); λg was the waveguide wavelength of the center frequency. The measurement of the fabrication BPF showed the center frequency of 3.25 GHz, the insertion loss $|S_{21}|$ less than 2.5 dB, and the return loss $|S_{11}|$ greater than 10.5 dB. Thus, by taking the cross-coupling due to source-load coupling of the feed line, the two transmission zero points were located at 2.65 GHz and 3.65 GHz on two sides of the passband, both of which were lower than 40 dB. Although the simulation of the designed filter was better than the measurement, the simulated result and the measured result still conformed to the proposed design concept. Typically, in order to minimize reflection characteristics, a tapered line of the I/O ports needs a length of 1/2 λg at a center frequency of 3.5 GHz, it is thus believed that the loss and reflection characteristics due to the short length of the tapered line resulted in deterioration of the performance of the entire filter. In this study, the use of traditional microstrip line structure formed on the metal surface above the SIW is studied and verified to have an effect on the entire resonant properties combined with the characteristics of the SIW cavity and the microstrip resonators. The design concept of the SIW filter with embedded IS can be used in the current future 5G communication system.

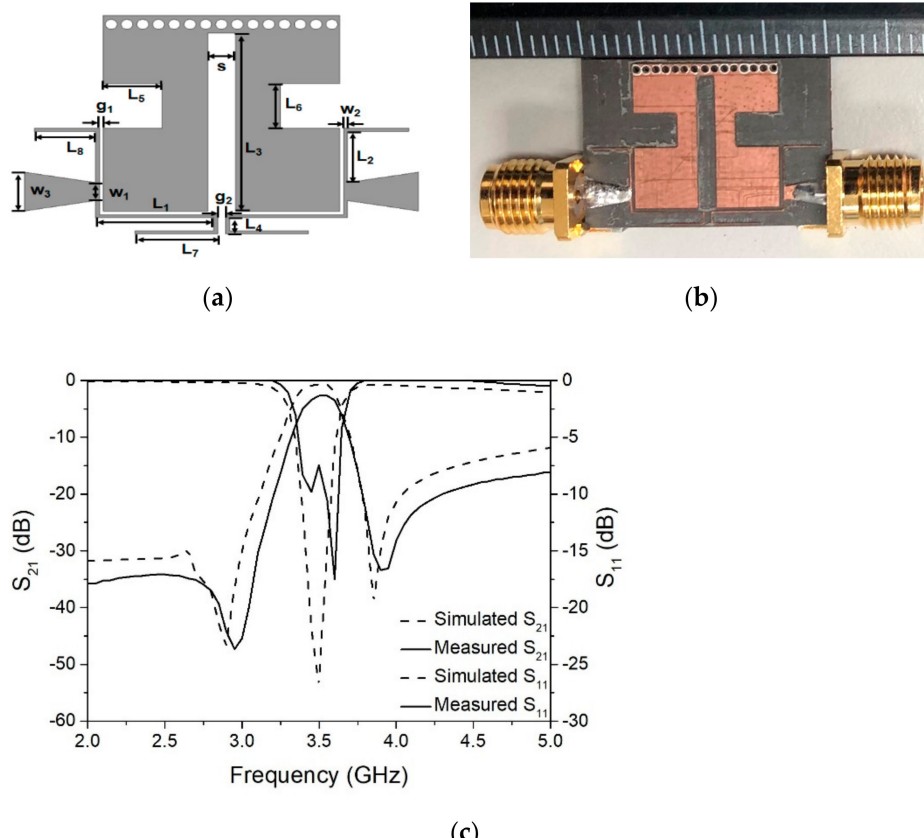

**Figure 8.** (**a**) Structure, (**b**) picture of fabricated sample, and (**c**) simulated and measured results of the fabricated BPF using HMSIW resonators with embedded SIS. ($L_1 = 6$, $L_2 = 4$, $L_3 = 11$, $L_4 = 1$, $L_5 = 4$, $L_6 = 2.77$, $L_7 = 5.11$, $L_9 = 5.3$, $g_1 = 0.2$, $g_2 = 0.4$, $w_1 = 1.11$, $w_2 = 0.2$, mm in all).

## 4. Conclusions

In this paper, we have shown that with the introduction of step impedance structure (SIS) on the top metal of the HMSIW resonator, the operating frequency of the filter can be reduced easily. The entire resonant properties are combined with the properties of the SIW cavity and the microstrip resonators, which can further be used to tune the frequency and the bandwidth of the BPF formed by the HMSIW resonator. The design concept is verified in this paper. The designed filter example which has the center frequency of 3.5 GHz is fabricated and measured, showing the insertion loss $|S_{21}|$ less than 3 dB, the return loss $|S_{11}|$ greater than 10 dB. The transmission zeros appeared at 2.95 GHz and 3.95 GHz, with attenuation of 45 dB and 35 dB respectively, due to the source-load cross-coupling structure. Due to the use of the half-mode, as well as the SIS, the proposed filter integrated with microstrip structure is smaller than the conventional SIW filter and has advantages of high-frequency selectivity and the band tuning ability. The advantage of the proposed design concept is that filter design using the proposed HMSIW in a certain frequency range becomes simple. When the BPF using HMSIW resonators without embedded SIS is designed first, center frequency adjustment can be easily obtained after embedding the SIS. The design concept of the SIW filter can be used in the current future 5G communication system.

**Author Contributions:** Conceptualization, M.-H.W.; methodology, M.-H.W. and C.-Y.T.; software, C.-Y.T.; validation, C.-Y.T. and Y.-C.C.; formal analysis, M.-H.W. and C.-Y.T.; investigation, C.-Y.T. and Y.-C.C.; resources, R.-Y.Y.; data curation, C.-Y.T. and D.-L.C.; writing—original draft preparation, M.-H.W. and C.-Y.T.; writing—review and editing, R.-Y.Y.; visualization, M.-H.W. and C.-Y.T.; supervision, R.-Y.Y.; project administration, R.-Y.Y.; funding acquisition, M.-H.W. All authors have read and agreed to the published version of the manuscript.

**Funding:** This study was funded by the Ministry of Science and Technology, Taiwan, R.O.C. under contracts MOST 109-2221-E-020-010. This work was also supported partly by the Putian University's Initiation Fee Project for Importing Talents for Scientific Research.

**Informed Consent Statement:** Not applicable.

**Acknowledgments:** The authors would like to thank the Precision Instrument Center of National Pingtung University of Science and Technology for supplying experimental equipment, and Hong-Zheng Lai and Shih-Kun Liu for the help with sample measurement.

**Conflicts of Interest:** The authors declare no conflict of interest.

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
