# Peer review of "A Bandpass Filter Using Half Mode SIW Structure with Step Impedance Resonator"

_electronics, doi:10.3390/electronics10010051_

Round 1

Reviewer 1 Report

The topic of this manuscript falls within the scope of this journal.

But this paper is not sufficient quality or novelty to be published in this journal.

1) There is a lengthy explanation of the SIW in the introduction, but a more detailed explanation of the actual HMSIW and the author's design is needed.

2) The author said that the size was reduced by simply inserting SIR, but no theoretical explanation is provided.

3) In the case of HMSIW without SIR shown in Figure 2, the size of [18] is used almost as it is (excluding L2). The authors only changed the TACONIC dielectric constant 3.5, 0.5mm thick substrate in Reference [18] to DUROID dielectric constant 2.2, 0.78mm substrate. This is believed to have seriously plagiarized [18] in the basic design. It is judged that this was not designed by myself, but merely changed the board to avoid plagiarism. The author made the mistake of claiming that the 50ohm line width is 1.11mm. Actual is 78 ohm. Section 2.1 is simply a parametric simulation of Ref. [18]. Looking at the design dimensions, you can see that the size has not been reduced at all.

4) In line 120, you need to change HMSIE to HMSIW.

On line 122, 4.5% GHz to 7% doesn't make sense. Either delete the GHz or mark the bandwidth in MHz.

5) The result of Fig. 3 according to the change of L3 in lines 123~125 is to be seen as an extension of the high band frequency rather than a change in the center frequency.

6) The explanation for Fig.9 is missing.

7) From 152 to 169, the author said he reduced the size. (Especially 159, 168) 30%, 50% at Fig7, and 35%, 55% at Fig8. However, no evidence or criteria are provided. Line 178 claims that the total size is 12x12mm, but since L3 alone is 11mm, the author is judged to have considered only the size of the resonator part of HMSIW excluding the feeder line and dielectric substrate. 24.8x20mm in Reference [18] is the total size excluding connectors. They must be compared under the same conditions. Obviously, judging from Fig. 10(a), the author's design appears to be approximately 21 x 12mm, but this is only due to the fact that the board was cut as short as possible.

8) There seems to be a lot of difference between the simulation and measurement results in Fig. 10(b). The insertion loss is also rather large. The author says the reason is a conductive silver glue problem, but I can't find any reasonable reason. Rather, it is judged to be the result because the feed line was not 50oHm. The reason why the simulation result is good is that if it is simulated with HFSS, it is automatically normalized to 78ohm when setting the waveport.

Overall, it followed the reference [18] as it was, and claimed to have reduced the size through SIR, but it did not reduce the size at all, and there is no improvement in performance or any difference in performance. Since it is not simply inserting the SIR, it is not considered enough to be published in this journal.

Author Response

The comments of the reviewers are very useful and constructive to the authors. We have tried our best to amend the manuscript to meet the requirement of the reviewer and goal of the journal. The revised part is marked with yellow color.

Reviewer 2 Report

The paper shows a miniaturized bandpass filter using half mode SIW
structure with step impedance resonator.

In my opinion authors failed to show the scientific novelty of the paper. It appears more an engineering work then a research paper.

Please better discuss the advance with respect to the known state of the art not only in terms of performances.

Author Response

(The authors gave the same response as above.)

Round 2

Reviewer 1 Report

The overall correction and additional explanation seems to have been done well. However, there are still some problems.

  1. From Reply 2~3, author says that "the purpose of this
    article is to prove that introducing microstrip type of SIS on the surface metal of the waveguide can effectively reduce the frequency of operation, thus reducing the size of the component", still the principle and explanation of the size reduction due to SIS are insufficient. 
  2. The focus is on easily changing the frequency by L5 and L6, and the overall size of the filter does not seem to be significantly reduced if compared under the same conditions even for designs without SIR. Therefore, the focus of this paper is on the simplicity of design and the ease of frequency change, but it is judged that the miniaturization of size should not be the main issue. Therefore, it is recommended to update the overall thesis structure design.
  3. The author said it was simply a mistake, but designed it with 78 ohms, and analyzing based on this is not an optimized result. And since the measurement equipment also has an input and output of 50 ohms, there are errors in various factors such as analysis and analysis with measurement. This result will be very confusing to the reader. Therefore, it is judged that creating a new prototype, analyzing, and measuring it, and updating the paper, will give the reader accurate information. It would be nice to consider the quality of the MDPI journal. Sorry for your inconvenience, but we recommend re-production and re-submission. It would be better to focus on the easy frequency tuning through SIR rather than miniaturization.(The difference between measurement and analysis is judged to be due to impedance mismatch when measuring a circuit designed with 78 ohms with a network analyzer rather than the problem presented by the author. This is because the author is judged to have calibrated to 50 ohms. It does not appear normalized to 78 ohms.)

Author Response

The overall correction and additional explanation seems to have been done well. However, there are still some problems.

  1. From Reply 2~3, author says that "the purpose of this
    article is to prove that introducing microstrip type of SIS on the surface metal of the waveguide can effectively reduce the frequency of operation, thus reducing the size of the component", still the principle and explanation of the size reduction due to SIS are insufficient. 

Reply: Thanks for the reviewer and the comment is well received. We have amended the description by “the purpose of this article is to prove that introducing microstrip type of SIS on the surface metal of the waveguide can effectively reduce the frequency of operation, thus tuning the resonant frequency to the desired passband without degrading the filter performance, thus reducing the size of the component",

  1. The focus is on easily changing the frequency by L5 and L6, and the overall size of the filter does not seem to be significantly reduced if compared under the same conditions even for designs without SIR. Therefore, the focus of this paper is on the simplicity of design and the ease of frequency change, but it is judged that the miniaturization of size should not be the main issue. Therefore, it is recommended to update the overall thesis structure design.

Reply:

Thanks for the reviewer and the comment is well received. We have deleted the description related to the “miniaturization of size “ through the overall paper and focused on the simplicity of design and the ease of frequency change.

  1. The author said it was simply a mistake, but designed it with 78 ohms, and analyzing based on this is not an optimized result. And since the measurement equipment also has an input and output of 50 ohms, there are errors in various factors such as analysis and analysis with measurement. This result will be very confusing to the reader. Therefore, it is judged that creating a new prototype, analyzing, and measuring it, and updating the paper, will give the reader accurate information. It would be nice to consider the quality of the MDPI journal. Sorry for your inconvenience, but we recommend re-production and re-submission. It would be better to focus on the easy frequency tuning through SIR rather than miniaturization.(The difference between measurement and analysis is judged to be due to impedance mismatch when measuring a circuit designed with 78 ohms with a network analyzer rather than the problem presented by the author. This is because the author is judged to have calibrated to 50 ohms. It does not appear normalized to 78 ohms.)

Reply:

Thanks for the reviewer and the comment is well received. To avoid the confusion to the reader, we have created a new prototype, measured it, and updated in the revised paper. we have designed the input and output port as the tapered structure with 78 Ω connected to the coupling feed line and with 50 Ω connected to the network analyzer, as shown in Figure 8 (a). However, the simulated and measured results still have a mismatch and we think it is caused from the fabrication error.

We have added the description “ Based on the above discussion and the design curve of Figure 6, a filter example is designed at 3.5 GHz to verify the design concept. Moreover, since the line impedance of the pair of microstrip coupling feed line is very high impedance, the input and output port are designed as the tapered structure with 78 Ω connected to the coupling feed line and with 50 Ω connected to the network analyzer. Figure 8 shows (a) structure, (b) picture of fabricated sample, and (c) simulated and measured results of the designed BPF example using HMSIW resonators with embedded SIS. “ and “The mismatch between simulation and measurement may be due to the fabrication error and shall be improved in the future.”

Reviewer 2 Report

The paper can now be accepted

Author Response

The paper can now be accepted

Reply: Thanks for the reviewer.

Round 3

Reviewer 1 Report

Thank you for the author's faithful response and the hard work of modifying the prototype.

Just a few more comments.

  1. In order to connect to devices designed with other 50 ohm systems, the input/output impedance must be 50 ohms, so if you redesign, you must design 50 ohms.
  2. Tapered line to change the input/output impedance of 78 ohms to 50 ohms, but it is a mistake. For example, for impedance conversion at 3500MHz, a quarter-wave transformer is usually used, or a tapered line needs a length of 1/2 guided wavelength at center frequency in order to minimize reflection characteristics. Therefore, 1/2 wavelength of this board is approximately 31mm. If this condition, it becomes a taper with minimal loss. Or, if you use a 1/4 wave transformer, you need 15.6mm length. If you use shorter than this design, insertion loss and return loss cannot be good. Another way is to design it for 78 ohms and calibrate the network analyzer to 78 ohms, but that would lead to an impractical design at all. This is because users will need a 50 ohm input/output device. Therefore, it is recommended to design 50 ohms from the initial design.
    It is judged that the loss and reflection characteristics due to the short length of the tapered line resulted in deterioration of the performance of the entire filter.
    If only this part is improved, it is considered a great paper.

Thank you for the work of the author.

Author Response

Thanks very much for the reviewer and this comment is well received. We have amended our discussion on the difference between the simulation and the measurement, and added the description as below:

In order to minimize reflection characteristics, a tapered line of the I/O ports needs a length of 1/2 guided wavelength at center frequency of 3.5 GHz it is believed that the loss and reflection characteristics due to the short length of the tapered line resulted in deterioration of the performance of the entire filter.
